# Fish Sidestream-Derived Protein Hydrolysates Suppress DSS-Induced Colitis by Modulating Intestinal Inflammation in Mice

**DOI:** 10.3390/md19060312

**Published:** 2021-05-28

**Authors:** Maria G. Daskalaki, Konstantinos Axarlis, Tone Aspevik, Michail Orfanakis, Ourania Kolliniati, Ioanna Lapi, Maria Tzardi, Eirini Dermitzaki, Maria Venihaki, Katerina Kousoulaki, Christos Tsatsanis

**Affiliations:** 1Laboratory of Clinical Chemistry, Medical School, University of Crete, 70013 Heraklion, Greece; m.daskalaki@med.uoc.gr (M.G.D.); konax@outlook.com (K.A.); orfanakis3012m@gmail.com (M.O.); raliakolliniatis21@gmail.com (O.K.); iwanna_lapi@hotmail.com (I.L.); renaderm@med.uoc.gr (E.D.); venihaki@med.uoc.gr (M.V.); 2Institute of Molecular Biology and Biotechnology, FORTH, 71100 Heraklion, Greece; 3Department of Nutrition and Feed Technology, Nofima AS, 5141 Bergen, Norway; Tone.Aspevik@Nofima.no (T.A.); katerina.Kousoulaki@Nofima.no (K.K.); 4Laboratory of Pathology, School of Medicine, University of Crete, 70013 Heraklion, Greece; tzardi@med.uoc.gr

**Keywords:** fish protein hydrolysates, colitis, cytokines, chemokines, IL-6, IL-10, inflammation

## Abstract

Inflammatory bowel disease is characterized by extensive intestinal inflammation, and therapies against the disease target suppression of the inflammatory cascade. Nutrition has been closely linked to the development and suppression of inflammatory bowel disease, which to a large extent is attributed to the complex immunomodulatory properties of nutrients. Diets containing fish have been suggested to promote health and suppress inflammatory diseases. Even though most of the health-promoting properties of fish-derived nutrients are attributed to fish oil, the potential health-promoting properties of fish protein have not been investigated. Fish sidestreams contain large amounts of proteins, currently unexploited, with potential anti-inflammatory properties, and may possess additional benefits through bioactive peptides and free amino acids. In this project, we utilized fish protein hydrolysates, based on mackerel and salmon heads and backbones, as well as flounder skin collagen. Mice fed with a diet supplemented with different fish sidestream-derived protein hydrolysates (5% *w/w*) were exposed to the model of DSS-induced colitis. The results show that dietary supplements containing protein hydrolysates from salmon heads suppressed chemically-induced colitis development as determined by colon length and pro-inflammatory cytokine production. To evaluate colitis severity, we measured the expression of different pro-inflammatory cytokines and chemokines and found that the same supplement suppressed the pro-inflammatory cytokines IL-6 and TNFα and the chemokines Cxcl1 and Ccl3. We also assessed the levels of the anti-inflammatory cytokines IL-10 and Tgfb and found that selected protein hydrolysates induced their expression. Our findings demonstrate that protein hydrolysates derived from fish sidestreams possess anti-inflammatory properties in the model of DSS-induced colitis, providing a novel underexplored source of health-promoting dietary supplements.

## 1. Introduction

Intestinal homeostasis is to a large extent maintained by a balance between the host immune system, the gut epithelium, dietary metabolites and, importantly, the gut microbiome [1]. Intestinal inflammation is the basis for a vast number of celiac and bowel pathologies, and manifests symptoms such as diarrhea, abnormal weight loss, fatigue, pain and malnutrition [2,3,4]. It is usually triggered by the combination of tissue damage and inciting environmental factors, such as gut microbes, but also dietary components that increase intestinal permeability [5,6].

Inflammatory bowel disease (IBD) consists mainly of Crohn’s disease (CD) and ulcerative colitis (UC). Although its exact etiology remains unclear, it is mainly characterized by an excessive inflammatory response in a genetically pre-disposed host with flare ups and relapses due to environmental factors such as antibiotic administration, microbial dysbiosis or dietary factors [7]. Chemically-induced colitis models are broadly utilized for the study of inflammatory disease pathogenesis because they successfully simulate human intestinal inflammatory diseases [8]. Specifically, dextran sodium sulfate (DSS)-induced colitis is a widely used animal model for the study of intestinal inflammation, as it is able to simulate acute, chronic and relapsing intestinal inflammation by modulating DSS concentration and administration frequency [9]. 

Standard treatment of intestinal inflammation includes immune suppressant or anti-inflammatory drug administration; yet, up to 40% of patients do not respond to a specific drug and there is still limited evidence that could be used to predict a patient’s response to a particular treatment [10]. In addition to therapy with such drugs, dietary interventions are an important component of combating IBD. The recommended first-line treatment for CD is exclusive enteral nutrition (EEN), which is a specialized, explicitly liquid nutritional formula destined for short-term consumption [11]. Other diets have also emerged, whose basic principle is the elimination of foods that exacerbate IBD symptoms, with the specific carbohydrate diet (SCD) and a diet low in fermentable oligo-, di- and monosaccharides and polyols (FODMAPs) being the most effective ones [12]. In any case, the connections between IBD and diet are vast, and patients are advised to follow a well-balanced diet, such as the Mediterranean, avoiding pitfalls of the Western and American diets, such as the high consumption of processed foods and vegetable oils or the poor intake of fruits and vegetables [13,14,15,16].

Additionally, dietary products that possess anti-inflammatory properties, and have been shown to alleviate inflammation in different disease settings, could possess a supportive role in the treatment of IBD. Indeed, nutrients and components derived from the diet can modulate intestinal inflammation by directly shaping the microbiota of the intestine and their excreted metabolites, and by exhibiting intricate immunoregulatory effects, including macrophage polarization [17,18,19,20]. Examples of such dietary compounds include vitamin D, fermentable fibers and curcumin [13]. Omega-3 polyunsaturated fatty acids (PUFAs), such as the plant α-linoleic acid and the fish eicosapentaenoic (EPA) and docosahexaenoic acid (DHA), are potent anti-inflammatory dietary constituents [21]. Interestingly, fish oil induces a powerful anti-inflammatory effect in the settings of intestinal inflammation and ameliorates colitis symptoms in animal models [22,23,24]. However, current evidence is insufficient to recommend intake of omega-3 fatty acids in clinical practice, although some beneficial trends can be noted [25,26,27].

The fish industry worldwide produces large quantities of sidestream biomass, which includes heads, gills, hearts, viscera, bones and roe and almost half of this is treated as waste material [28]. During recent years, conventional and alternative “green” extraction methods allowed for the valorization of fish sidestreams by extracting high-value compounds including fish oils such as omega-3 EPA and DHA [29]. Although the majority of fish-derived nutritional supplements consist of fish oil and are extracted from tissues of oily fish, there are many other highly nutritional biologically active compounds found in fish sidestreams, such as vitamins, collagen, chitin, minerals and polyunsaturated fatty acids (PUFAs), as well as high amounts of protein [30,31,32,33]. Although most often discarded, fish sidestream proteins, especially bioactive peptides, may have important beneficial properties, which are currently underexplored. In this study, selected nutritional supplements prepared from hydrolyzed fish-derived sidestream proteins [34] were tested in vivo for their potential to suppress inflammation, in the murine model of acute intestinal inflammation induced by DSS.

## 2. Results and Discussion

### 2.1. Fish-Derived Protein Hydrolysates Partly Suppress DSS-Induced Colitis Development

To investigate the potential anti-inflammatory effect of fish sidestream-derived protein hydrolysates (Table 1) in acute intestinal inflammation, we used the murine DDS-induced colitis model, which is broadly used as a model of inflammatory bowel disease. The composition of the different supplements and their organoleptic properties have been recently described and are presented in Appendix A [34]. We introduced six female mice per group to a diet supplemented with 5% w/w of the respective extract for 10 days to allow mice to become accustomed to the new diet. Soy protein at a concentration of 5% was used as control to simulate similar protein intake, as fish-derived supplements contained a substantial amount of protein and because soy is the predominant source of protein in the control murine diet. We then proceeded to the intestinal inflammation protocol according to which 3% w/v DSS was administrated in drinking water for five consecutive days, followed by a three-day recovery period [35]. Mice were weighed daily and simultaneously disease progression was scored (Figure 1A).

DSS is a chemical toxin which causes tissue damage by disrupting the epithelial monolayer lining of the intestine. As a result, pro-inflammatory luminal content, consisting of microorganisms and their metabolites, among others, ingress in the underlying tissue, leading to an exacerbated immune response and inflammation, central mediators of which are intestinal macrophages [9,36,37]. Macrophages also play a key role in the subsequent termination of inflammation and the healing of the intestine [24]. Major parameters reflecting DSS-induced intestinal inflammation severity include body and spleen weight, colon length, diarrhea and rectal bleeding [38]. 

As expected, the colon of DSS-treated mice significantly contracted due to inflammation and tissue damage in comparison to the non-DSS control group (Figure 1B,C). Notably, the HSH nutritional supplement diet group exhibited significantly less colon contraction compared to the DSS control group, suggesting that the HSH supplement administration suppresses intestinal inflammation and tissue damage. In order to further support our data, histological evaluation of DSS-induced tissue damage was performed using hematoxylin and eosin staining (H&E) in colon tissue sections (Figure 1D). As expected, the DSS-treated control group exhibited extensive tissue damage, cell necrosis and loss of crypts (asterisk) compared to untreated mice. Notably, HMB, HSB and HSH diet groups exhibited reduced crypt loss, whereas HSH + C, HMH and Collagen groups displayed areas of broad inflammation triggering tissue damage (asterisks) as well as inflammatory cells infiltration (arrow) (Figure 1D). In order to further assess DSS-induced tissue damage, blinded histological scoring was performed according to Chassaing et al. [38] (Figure 1E). In accordance with the colon length measurements, HSH supplementation reduced DSS-induced tissue damage, and in addition HSB and HSH + C significantly improved colon histological features post DSS treatment (Figure 1E). Spleen is an organ with important hematopoietic and immune functions, and it is known to expand during inflammation due to proliferation of inflammatory cells, a phenomenon termed as splenomegaly [39]. The results show that there was a clear tendency of the HSH diet group towards limited spleen enlargement (*p* = 0.07), even though no statistical significance was reached (Figure 1F,G). This finding, combined with the previous observations, indicates that, among all compounds tested, the HSH dietary supplement exhibited the most potent result, which suggests a strong anti-inflammatory effect that partly suppresses DSS-induced colitis in mice.

DSS administration rapidly damages the mucosal barrier of the intestine, leading to inflammation and significant weight loss. None of the supplements hindered weight loss (Figure 2). On the contrary, two nutritional supplement groups, the HMB and HMH, exhibited accelerated weight loss (Figure 2A,C), which may be due to a lower tolerance to mackerel sidestream-derived diets [34]. Notably, among the different experimental groups, the group receiving HMH protein hydrolysates as supplement showed higher water consumption compared to the control group (6.77 mL +/− 0.73 SD vs. 5.24 mL +/− 0.42 SD per mouse, respectively), which possibly explains the increased weight loss and some of the results presented below (Appendix A). The remaining groups exhibited similar water consumption to that of the control group, suggesting that the observed effects were attributed to the extracts’ properties and not to differences in DSS-containing water intake. Food intake did not differ between the different groups studied (Appendix A).

Throughout the experiment (5 days of DSS treatment and 3 days of recovery), colitis severity was measured daily based on a disease scoring system ranging from 0 to 3, as previously described [38]. During the first 5 days, no significant improvement was observed in any of the diet groups (Figure 3). In fact, four out of the six diet groups (HSB, HMH, HSH, and HMB) exhibited a higher disease score on day 4 compared to the DSS-control group. However, from day 6 to day 8, during which DSS treatment was terminated, no significant change in the recovery process was noted, and mice recovered from the worsening observed on day 4 in some groups (Figure 3). Nevertheless, mice recovered fast, suggesting that the supplements were not harmful to the intestine.

### 2.2. Nutritional Supplements Modulated Intestinal Inflammation through Cytokine and Chemokine Regulation 

The tolerogenic state of a healthy intestine is facilitated through a complex network of interactions, which includes a variety of immune cells, such as dendritic cells, macrophages and intraepithelial lymphocytes, as well as a range of important immune mediators, including IL-4, IL-10 and TGFb [40]. Tissue-resident macrophages of a healthy gut may exhibit an anti-inflammatory role by producing anti-inflammatory cytokines, such as IL-10, and therefore may contribute to the counterbalance of inflammatory events and the maintenance of colon homeostasis [41,42]. However, in colitis, due to mucosal barrier disruption, activated monocytes/macrophages infiltrate the colon, resulting in the secretion of pro-inflammatory cytokines, such as TNF-α, IL-1b and IL-6, orchestrating the inflammatory response to tissue damage [43]. To assess the possible anti-inflammatory actions of the tested supplements, tissue samples were collected, and the expression of major pro-inflammatory cytokines was measured.

Indeed, the expression of the aforementioned cytokines significantly increased in the intestines of DDS-treated mice compared to control (Figure 4A–C). Mice that received HSH as a nutritional supplement, which exhibited the least contracted colon and least expanded spleen (Figure 1), also expressed reduced *Tnf-α* mRNA, the hallmark cytokine of colitis and a therapeutic target, compared to the DSS-treated control group (Figure 4A). In addition, the same group exhibited reduced mRNA levels of *Il-6* in colon tissue (Figure 4B), an observation also confirmed at the protein level in intestinal tissue homogenates and serum (Figure 5A,B). TNF-α protein levels in colon and serum were not detectable, possibly due to the short half-life of TNFα and the recovery period following termination of DSS administration, during which the healing process had initiated. Interestingly, when the HSH supplement was combined with collagen, the beneficial effect was not evident (Figure 4A,B). This may be attributed to the reduction in the net amount of HSH by half, which may not be enough to exhibit any anti-inflammatory action. Although the HMB and HSB diet groups did not affect colon length, thereby suggesting that they may not effectively suppress inflammation and tissue damage, significant reduction in *Il-6* expression was observed (Figure 4B). The latter was also confirmed at the protein level in tissue homogenates and serum from mice that received the HSB nutritional supplement (Figure 5A,B). No differences in *Il-1β* expression were observed in any of the DSS-treated groups (Figure 4C). 

Chemokines are secreted factors that act as chemoattractants, inducing the migration of leukocytes from the blood stream to the site of inflammation. Intestinal inflammatory diseases, such as IBD, are characterized by the infiltration of neutrophils, monocytes and lymphocytes; therefore, chemokines have been proposed as critical modulators of intestinal inflammation [44]. Expression levels of two subfamily members of chemokines, CC and CXC, were measured in colon samples (Figure 6). CXCL1 chemokine, also known as KC, Groα and Gro1 oncogene, and CXCL2 chemokine, also termed as MIP-2, are members of the CXC chemokine subfamily and share approximately 90% similarity. They are expressed in macrophages, neutrophils and epithelial cells, are highly expressed in inflamed colon tissue and their main role is the recruitment of neutrophils to the infection site acting through the CXCR2 receptor [45]. CXCR2 knockout mice have been shown to present modest DSS-induced intestinal inflammation with reduced neutrophil recruitment and reduced kidney injury [46,47]. Interestingly, the HSH and HSH + C nutritional supplements as well as the HSB marginally downregulated the expression of *Cxcl1* mRNA, potentially contributing to a reduced inflammatory response (Figure 6A). When CXCL1 protein levels were measured, the HSB group had indeed significant reduction in CXCL1, while the HSH group also presented a strong tendency towards reduced CXCL1 levels (Figure 6C). On the contrary, the HMH (which consumed more DSS-containing water) and collagen-administered groups exhibited elevated expression levels of both *Cxcl1* and *Cxcl2* chemokine mRNA, indicating a possible exacerbation of the inflammatory response through neutrophil recruitment (Figure 6A,B). 

Next, we aimed to investigate the expression levels of two members of the CC chemokine subfamily that have been also found to play a significant role in intestinal inflammation and disease severity. CCL2 chemokine induces monocyte, T cell and dendritic cell migration to inflammatory sites through CCR2 and CCR4 receptors. It has been shown that CCR2^-/-^ mice exhibit lower eosinophilic inflammation and DSS-induced tissue damage [48], whereas CCL2 blockade results in decreased intestinal inflammation and colon tumorigenesis associated with chronic inflammation [49]. CCL2^-/-^ mice exhibit reduced colitis and mortality rate associated with restricted macrophage infiltration to the colonic mucosa [50]. In accordance with the previous data, the HSH nutritional supplement significantly lowered the expression of *Ccl2* chemokine, further reinforcing its anti-inflammatory potential (Figure 6C). Dietary supplements HSB and HMB marginally downregulated *Ccl2* expression in line with their anti-inflammatory potential observed in Figure 4B. 

CCL3 chemokine, also known as MIP-1α, has also been associated with macrophage recruitment as well as with granulocyte recruitment. It is involved in acute inflammation as it acquires inflammatory, pyrogenic and chemokinetic properties through CCR1, CCR4 and CCR5 receptors [45]. As a consequence, CCL3^-/-^ and CCR5^-/-^ mice exhibit reduced DSS-induced tumorigenesis [51], whereas CCR5-specific blockade attenuates intestinal inflammation by dysregulated trafficking of both innate and adaptive immune cells [52]. The HMH and collagen supplements increased the expression of *Ccl3* mRNA, whereas the remaining supplements did not affect *Ccl3* expression levels (Figure 6D). Overall, although the fish sidestream-derived supplements HSB, HMB, HSH and HSH + C reduced expression of the pro-inflammatory chemokines *Cxcl1* and/or *Ccl2*, the HMH and collagen supplements were found to upregulate pro-inflammatory chemokines (Figure 6); yet, no significant differences were observed in disease progression or the macroscopic phenotype of colitis (Figure 1). 

IL-10 is an anti-inflammatory cytokine and a major regulator of intestinal homeostasis, considering that the microbiota are in constant interaction with host immune systems, triggering immune activation [53]. In that context, IL-10 is necessary to establish an anti-inflammatory environment supporting microbial symbiosis. Intestinal inflammation disrupts this balance through the induction of pro-inflammatory cytokines and chemokines. In our study, acute intestinal inflammation was induced for five days through DSS administration and a subsequent three-day recovery period enabled us to examine the emergence of repair and tissue-healing mechanisms. Therefore, we measured *Il-10* and *Tgfb* expression in tissue samples. TGFb is a key regulator of the intestinal immunity by inducing differentiation and activation of regulatory T cells (Tregs), promoting epithelial cell proliferation and supporting tissue repair [54]. Both the HMH and collagen dietary supplements significantly upregulated the expression of these anti-inflammatory mediators (Figure 7), possibly due to an increased leukocyte infiltration, also supported by the increased chemokine expression shown in Figure 6, and/or to compensate for a more excessive inflammation. In addition, tissues from the HSB and HMB diet groups exhibited increased expression of *Tgfb* (Figure 7B), further confirming their anti-inflammatory potential.

A mechanism to reduce inflammation would be through direct modulation of pro- or anti-inflammatory gene expression (i.e., macrophage polarization) [35,55]. Another mechanism would be to block or hinder leukocyte—primarily macrophage—infiltration to the inflamed tissue. It is not clear by which of the aforementioned ways the observed effects of the supplements were mediated, since macrophage infiltration was not quantified. For example, it could be possible that the HMH and collagen groups do not show a reduction in the expression of the pro-inflammatory markers because the latter is masked by a larger macrophage infiltration. 

Overall, some of the tested dietary supplements could have a beneficial role in the alleviation of DSS-induced colitis symptoms caused by severe inflammation, as is also seen in other studies with either DSS [56] or TNBS-induced colitis [57], using protein hydrolysates of plant origin. The HSH supplement stands out as having the most potent anti-inflammatory effect, which can be shown by the healthier colon length, tissue integrity shown by histology and spleen weight, by the lower pro-inflammatory gene expression and the reduction in pro-inflammatory chemokines. The HSB nutritional supplement had also an anti-inflammatory potential, since it exhibited a tendency for reduced levels of inflammatory markers, that is, lower *Il-6* and elevated *Tgfb* expression, as well as reduced tissue damage when examining colon samples histologically. On the other hand, the HMH and collagen supplements might exacerbate inflammation, which is not unusual for diets rich in animal proteins [58], although no significant differences were observed in colon length and spleen weight. It should be noted, however, that the case of the HMH supplement could probably be explained by the higher consumption of water containing DSS. 

The exact mechanism through which the examined supplements act in the above settings of acute intestinal inflammation would be immensely difficult to define, and is beyond the scope of this study. The former is justified by the tremendously complex interplay between dietary components, host immunity, the intestinal barrier function and the gut microbiota. It is highly likely that the tested protein hydrolysates of fish origin modulated the composition of the gut microbiome, considering that animals were exposed to the supplements for a total of 15 days (7 days prior to the experiment plus 8 days of the experiment) and the fact that intestinal microbiota may respond promptly to modified diets [59]. In agreement with our hypothesis, tuna-derived fish supplements were shown to ameliorate DSS-induced intestinal inflammation through short-chain fatty acid production and microbiome modulation [60]. We have also recently shown that different protein-containing dietary supplements affect the gut microbiome [61].

The tested supplements consist primarily of protein but also a fraction of minerals. Lipids were actively removed and a remnant of 1–2% was present, thus their potential contribution in the anti-inflammatory action is rather limited. The contribution of minerals cannot be excluded, since minerals such as Zn possess anti-inflammatory actions. Nevertheless, the major content of the supplements is proteins consisting of a wide range of amino acids. It has been exhibited that intestinal inflammation alters amino acid metabolism both in the immune and non-immune cells of IBD patients and their intestinal microbiome, thus creating a shift in nutritional demands and utilization [62,63]. Possibly, each amino acid has a distinct part in intestinal inflammation, with some exacerbating it and others suppressing it [16,64,65]. For example, dietary glycine protected rats from chemically-induced colitis by abrogating the production of pro-inflammatory cytokines and chemokines [66]. Interestingly, the HSH supplement, which had the most beneficial effect of the examined extracts, is rich in glycine, as are the HSB and HMB extracts, which also presented some anti-inflammatory action (Appendix A). Notably, collagen has the highest glycine concentration, yet it presented an increase in both anti-inflammatory and pro-inflammatory markers; thereby it would be over-simplistic to attribute the observed effects solely to a specific amino acid. Instead, the entire amino acid profile of each extract and the potential action of bioactive peptides should be accounted for. 

Moreover, DSS-induced colitis in rats was alleviated when supplemented with threonine, serine, proline and cysteine, in part through stimulation of mucin production and equilibration of the gut microbiome [67]. With the exception of cysteine, the HSH, HSB and HMB supplements contain considerable levels of the aforementioned amino acids. Histidine downregulated the production of pro-inflammatory markers by macrophages when offered in mice with colitis [68]. Arginine, which is found in important concentrations in the examined fish extracts, reduced intestinal inflammation in rats and mice with chemically-induced colitis [69,70,71]. Overall, the fish-derived protein hydrolysates that were tested in this study contain considerable quantities of amino acids that have been demonstrated to ameliorate symptoms of colitis, through modulation of amino acid metabolic pathways. Taking everything into account, we propose that fish sidestream-derived protein hydrolysates may contribute to the alleviation of intestinal inflammation and support intestinal homeostasis.

## 3. Materials and Methods

### 3.1. Materials

Enzymatic protein hydrolysates based on mackerel heads (HMH), mackerel backbones (HMB), salmon heads (HSH) and salmon backbones (HSB) were produced according to Aspevik et al. (2021) [15] in the pilot raw material processing plant of Nofima AQUAFEED Technology Center (Fyllingsdalen, Norway). Flounder skin collagen was kindly provided by Seagarden (Karmøy, Norway). Chemical properties of protein hydrolysates based on mackerel and salmon are shown in Appendix A and in Aspevik et al. (2021) [15]. Chemical properties of protein hydrolysate of collagen are shown in Appendix A. All other chemicals were of analytical grade.

### 3.2. Animal Maintenance

Animal housing, handling and all procedures were according to national and EU legislation on laboratory animal handling and approved by the University of Crete Ethics Committee (license number 269904). C57BL/6 mice were maintained in a 12 h day/night cycle and 21–23 °C conditions prior to treatment in a pathogen-free animal facility in the Medical School of the University of Crete, Heraklion, Greece.

### 3.3. DSS-Induced Colitis

Six female C57BL/6 mice, 6–8 weeks of age, were fed a normal chow diet (4RF21, Mucedola, Settimo Milanese, MI, Italy) plus 5% *w*/*w* fish sidestream-derived protein hydrolysates for one week prior to the experiment initiation day or purified soy protein as control. The control group was supplemented with purified soy protein throughout the experiment. Then, they were treated with 3% *w*/*v* DSS (40kDa, A3261 Applichem, Darmstadt, Germany) in the drinking water for 5 days to initiate the inflammatory phenotype and then left to recover for 3 days, before sacrifice. Disease score and animal weight were monitored daily. Disease score was measured by placing every mouse in an empty cage for 15 min where feces were collected and scored as follows: normal stool consistency—0, soft stools—1, very soft stools with traces of blood—2 and watery stools with visible rectal bleeding—3. Spleen, colon and blood serum were collected at the time of animal sacrifice, and spleen weight and colon length were measured. Samples were stored in −80 °C for further analysis. 

### 3.4. RNA Isolation and Quantitative PCR

Colon tissue was homogenized using a mechanical homogenizer in TrizolTM reagent (15596-026, ThermoFisher, Waltham, MA, USA) and total RNA was extracted according to manufacturer’s instructions. Then, 5004 ng of total RNA was reverse transcribed using a PrimeScript™ RT reagent Kit (Perfect Real Time) (RR037A, TaKaRa, Bio Inc, Kusatsu, Shiga, Japan). Each sample was diluted five times and used as a template in duplicates for two-step quantitative PCR reactions in a 7500 Fast Real-Time PCR Instrument (Applied Biosystems^®^, Foster City, CA, USA) with 96-well Block Module as follows: start step 95 °C for 3 min, and then 40 cycles of 95 °C for 10 s and 60 °C for 30 s followed by melting curve. Amplification was performed using a KAPA SyBr^®^ Fast Universal qPCR kit (KK4618,Kapa Biosystems, Wilmington, MA, USA) The primers used are listed in Appendix A. Data analysis was performed using mRNA levels expressed as relative quantification (RQ) values, which were calculated as RQ = 2(-DDCt), where DCt is (Ct (gene of interest)—Ct (housekeeping gene)). Actin mRNA was used as the internal control gene.

### 3.5. ELISA

Colon tissue was homogenized using a mechanical homogenizer in 1x PBS supplemented with CompleteTM Inhibitor Cocktail (11697498001, Merck, Kenilworth, NJ, USA) protease inhibitors. Samples were centrifuged and supernatants were used for ELISA assay. Blood serum was diluted 5 times in 1x assay diluent provided by ELISA kit and used for cytokine measurement. Cytokine concentration of mouse IL-6 (431301, ELISA Max™ Delux Set BioLegent, San Diego, CA, USA) and Cxcl1 (Mouse CXCL1/KC DuoSet Elisa, DY-453, R&D Systems, Minneapolis, MN, USA) was measured according to manufacturer’s instruction. 

### 3.6. Histological Analysis

For histopathological analysis, the large intestine of the mouse was removed, fixed in buffered formalin and embedded in paraffin. Then, sections of 5μM were prepared, placed on glass lesions and stained with hematoxylin and eosin (H&E) to assess inflammatory cell infiltration and tissue damage. Blinded histological colon tissue scoring was performed according to Chassaing et al. [38] based on epithelial tissue damage and inflammatory infiltration into the mucosa, submucosa and muscularis/serosa, resulting in a total score of 0–36 points per mouse.

### 3.7. Statistical Analysis

Data shown in Figure 1, Figure 4, Figure 5, Figure 6 and Figure 7 are shown in boxes min to max with median ± SD, and the ones in Figure 2 and Figure 3 are presented as mean ± SD. Statistical analysis was performed using Graphpad Prism 7.0 (GraphPad Software, San Diego, CA, USA). A Mann–Whitney t-test was performed to test statistical significance of each diet group to DSS control diet. Tukey’s test was performed to test statistical significance between groups, confirming the results. Differences with a *p* value < 0.05 are considered significant (* indicates *p* < 0.05, ** indicates *p* < 0.01, *** indicates *p* < 0.001).

## 4. Conclusions

Our findings demonstrate for the first time that fish sidestream-derived protein hydrolysates can suppress chemically-induced colitis. Recent evidence supports the health-promoting properties of bioactive peptides. The present report provides evidence for a new, underexplored source of dietary products with health-promoting properties.

## Figures and Tables

**Figure 1 marinedrugs-19-00312-f001:**
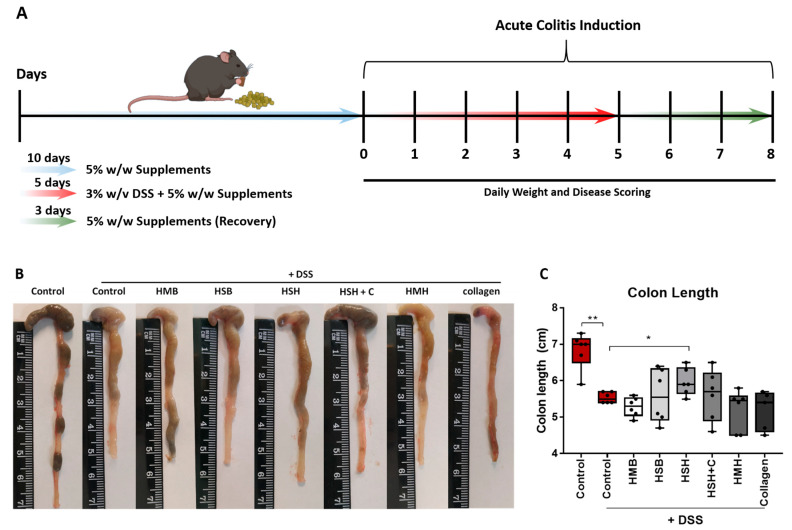
The effect of dietary supplements on colon length and spleen weight of mice with DSS-induced colitis. (**A**) Graphical illustration of the experimental design. (**B**) Representative intestines from each diet group. (**C**) Length measurements of the total intestines. (**D**) Hematoxylin and eosin tissue staining of colon tissue sections. White asterisks indicate tissue damage and loss of crypts; white arrow indicates infiltration of inflammatory cells. (**E**) Blinded histological scoring performed on H&E-stained colonic tissue. (**F**) Representative spleens from each diet group. (**G**) Weight measurements of the total spleens normalized to total body weight. Graphs represent median ± SD and an unpaired t-test was performed. * *p* < 0.05, ** *p* < 0.01, *** *p* < 0.001.

**Figure 2 marinedrugs-19-00312-f002:**
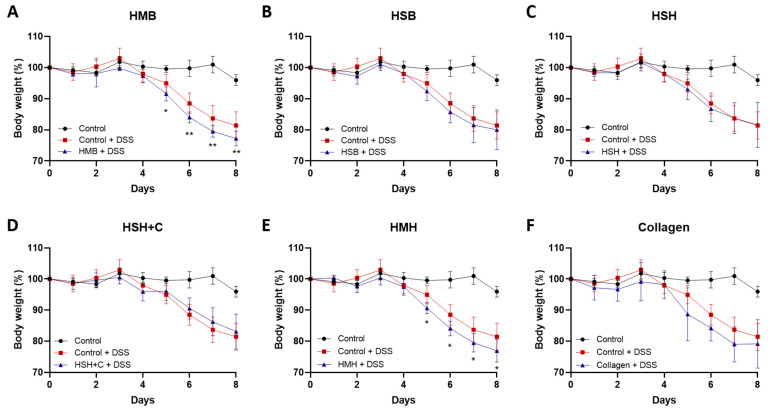
Body weight change of mice exhibiting DSS-induced colitis. (**A**–**F**). Total body weight was monitored daily and expressed as percentage of initial weight prior to experimental start in each diet group. Graphs represent median ± SD and 2-way ANOVA statistical analysis was performed. * *p* < 0.05, ** *p* < 0.01.

**Figure 3 marinedrugs-19-00312-f003:**
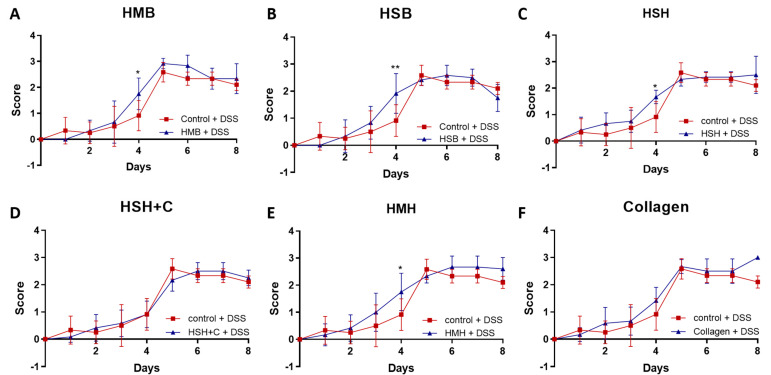
DSS-induced colitis disease progression. (**A**–**F**) Disease severity in each diet group was monitored daily and scored as follows: Score = 0: normal stools. Score = 1: soft stools with positive hemoccult. Score = 2: very soft stools with traces of blood. Score = 3: watery stools with visible rectal bleeding. Graphs represent median ± SD and 2-way ANOVA statistical analysis was performed. * *p* < 0.05, ** *p* < 0.01.

**Figure 4 marinedrugs-19-00312-f004:**
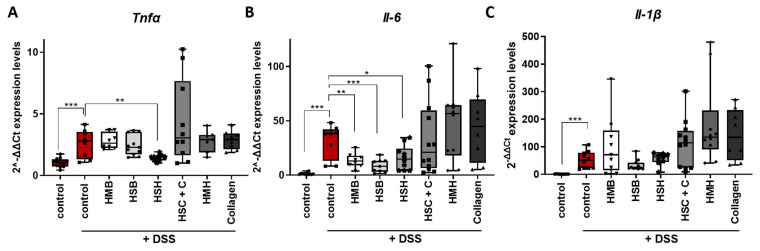
The effect of dietary supplements on mRNA levels of pro-inflammatory cytokines in colon tissue of DSS-treated mice. The expression profile of (**A**) *Tnfa*, (**B**) *Il-6* and (**C**) *Il-1β* was measured using real-time PCR in colon tissue. Graphs represent median ± SD and an unpaired t-test was performed. * *p* < 0.05, ** *p* < 0.01, *** *p* < 0.001.

**Figure 5 marinedrugs-19-00312-f005:**
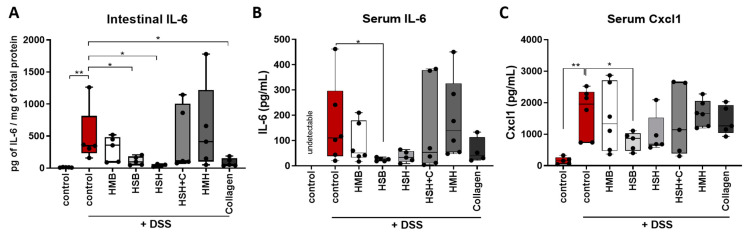
The effect of dietary supplements on protein levels of (**A**) intestinal IL-6, (**B**) serum IL-6 and (**C**) serum Cxcl1 quantified using ELISA. Graphs represent median ± SD and an unpaired t-test was performed. * *p* < 0.05, ** *p* < 0.01.

**Figure 6 marinedrugs-19-00312-f006:**
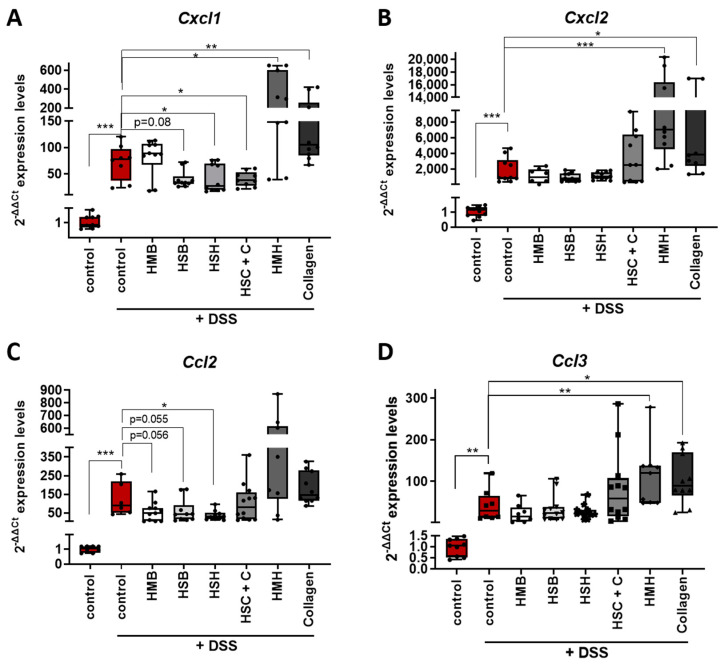
Evaluation of the expression of selected chemokines in DSS-induced colitis in mice. mRNA expression of (**A**) *Cxcl1*, (**B**) *Cxcl2*, (**C**) *Ccl2* and (**D**) *Ccl3* was quantified using real-time PCR in colon tissue of the selected diet groups. Graphs represent median ± SD and an unpaired t-test was performed. * *p* < 0.05, ** *p* < 0.01, *** *p* < 0.001.

**Figure 7 marinedrugs-19-00312-f007:**
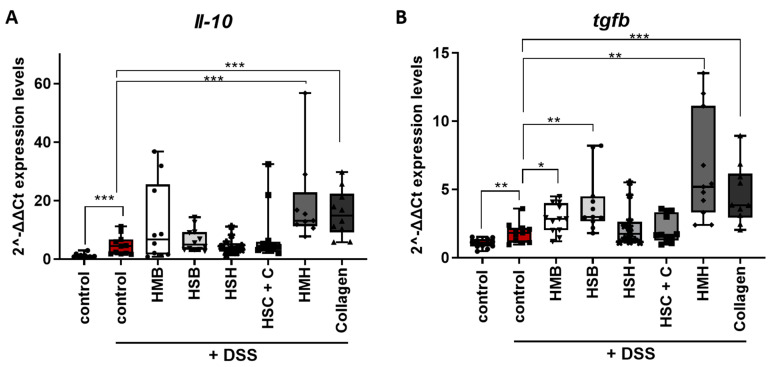
Monitoring the effect of dietary supplements on the expression of anti-inflammatory mediators in colon tissue of DSS treated mice. (**A**) *Il-10* and (**B**) *Tgfb* mRNA expression levels were measured using real-time PCR in the selected diet groups. Graphs represent median ± SD and an unpaired t-test was performed. * *p* < 0.05, ** *p* < 0.01, *** *p* < 0.001.

**Table 1 marinedrugs-19-00312-t001:** List of tested supplements and their composition.

Supplement	Composition
HMB HSB HSH HMH HSH + C	Hydrolysate Mackerel Backbone
	Hydrolysate Salmon Backbone
	Hydrolysate Salmon Heads
	Hydrolysate Mackerel Heads
	50% Hydrolysate Salmon Heads + 50% Collagen
Collagen	100% Collagen

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
