# Peer review of "Fish Sidestream-Derived Protein Hydrolysates Suppress DSS-Induced Colitis by Modulating Intestinal Inflammation in Mice"

_marinedrugs, 2021, doi:10.3390/md19060312_

Round 1

Reviewer 1 Report

This paper explores fish processing side-streams (waste to be processed usually into fish oil). The researchers used certain supplements (fish heads and fish bones) on laboratory mice, and then weighed/measured their spleens and colons.

The methodology section being placed after the results is confusing.

Each supplement was tested on 6 female mice. The 6 different supplements were hydrolysates of mackerel backbone, salmon backbone, salmon heads, mackerel heads, a mixture of salmon heads with collagen, and 100% collagen.

The mice are accustomed to this meal for 10 days, then are introduced to the chemical DSS  for 5 days to induce acute colitis; afterwards they are fed their meal for 3 more days to recover before data collection.

An ELISA kit was also used for cytokine measurement.

There seems to be a significant difference in both colon size as well as the cytokine measurements.

It was unclear if the organs (colon) were cleaned of materials before weighing.   Authors please explain ?

Why were there no spleen effects ?

Author Response

Comment: This paper explores fish processing side-streams (waste to be processed usually into fish oil). The researchers used certain supplements (fish heads and fish bones) on laboratory mice, and then weighed/measured their spleens and colons.

Comment 1.1: The methodology section being placed after the results is confusing.

Response 1.1: We thank the reviewer for the comment but the structure of the manuscript was according to the journal requirements.

Comment 1.2: It was unclear if the organs (colon) were cleaned of materials before weighing.   Authors please explain ?

Response 1.2: We apologize to the reviewer for the confusion. Concerning organ weight measurements, only spleens were weighted as spleen enlargement is the result of DSS-induced inflammation. Spleens were cleaned before weighing. Colon samples were used to measure their length as colon shortening reflects inflammation and has been widely used as a measure of DSS-induced colitis severity.

Comment 1.3: Why were there no spleen effects?

Response 1.3 : We thank the reviewer for the comment. Spleen size is an indirect indication of inflammation due to rapid leukocyte proliferation. Our results showed that mice treated with DSS, which causes an acute inflammatory response, have larger spleens (a phenomenon termed as “splenomegaly”) compared to the mice that were not exposed to DSS. As far as the effect of the tested supplements on spleen size is concerned, the HSH extract resulted in marginally significantly (p=0.07) smaller spleens compared to the control group, suggesting an anti-inflammatory action (Figure 1F) . Since it is an indirect measure of inflammation, the impact of the supplements was not as profound as in other measures. We have now included the p value in the results section in addition to the figure (line 43).

Reviewer 2 Report

This study evaluates the immunomodulatory properties of fish-derived proteins in the context of DSS-induced colitis. It is an interesting topic, but there are two main limitations of this study (detailed among the major points below):

  • DSS colitis is induced by adding DSS in the drinkingwater, thus severity of the colitis is intimely linked to the amount of water ingested by mice. At two times in the manuscript, the authors hypothesized that maybe their supplements increase salt concentration in mice food, and thus potentially increase their water consumption. If it is the case, we cannot conclude on the results of this study. As mentionned below, it is important that the authors evaluate the effects of their supplements on daily food and water intake, since these parameters have major effects on colitis severity.
  • Some classical readout of colitis are missing: histological analysis, intestinal permeability assay (dextran FITC) and more than one protein marker for evaluating colitis (only IL-6 in this study). All the study is based on the measure of mRNA level duringthe recovery period, it is informative but not totally relevant since transcriptional regulation of cytokines/chemokines is a higlhy dynamic process that might be complex to interpret.

Major points:

1) Introduction is very short, it might be of interest for the readers to more the already known effects of dietary interventions in the context of IBDs. Notably, is there some studies describing the effects of fish-based supplement (as evoked in the abstract by the authors :” Diets containing fish are known to suppress inflammatory diseases.”and/or based on proteins hydrolysates. The paragraph between lines 70 (the fish industry…) and 79 (protein) should be move in the introduction section (it does not contain results, and proposes interesting introductive elements).

2) Line 69: “Fish-derived protein hydrolysates partly suppresses inflammatory bowel disease development.”. Here (and throughout the manuscript) , if the authors refer to their model of DSS-induced colitis, they have to say “suppresses colitis” or “suppresses chemically-induced colitis” or “DSS-induced colitis” but not inflammatory bowel disease that refer to the human disease. This is confusing. Please edit here and throughout the manuscript.

3) Diet supplemented with fish-derived products might be consumed differently by mice since they have not the same organoleptic properties than control diet. Is there a control or measure of daily food intake between groups? Because, if there are some differences in the daily food intake between groups it can impact the response to colitis (for instance, caloric restriction is associated with anti-inflammatory effects). If available these data should be provided, if not, this possible bias should be mentioned by the authors.

4) Line 129-130 and Line 143-144: “or due to the higher salt content of the aforementioned supplements which could lead mice to a slightly higher DSS-containing water consumption” It might represent an important bias, ideally daily water drinking should be measured to evaluate the importance of this bias (alternatively another colitis model that not relies on drinking water should be used (TNBS for instance)).

5) Scoring system used in figure 3 is quite superficial and not very discriminant between groups, it is a pity that there is no histological scoring performed on paraffin embedded colon + HE staining (as usually performed in this kind of experiment).

6) Except for IL-6 (presented in figure S1) all colitis hallmarks were measured at transcriptional level. Since these measures have been done during the recovery period and only based on transcripts, it might be complex to conclude on the results obtained.

7) As noticed for the introduction, the (final) discussion section is very short too, it might be of interest to develop a bit more on the potential mechanism involved to explain the immunomodulatory properties of these protein hydrolysates. Effects on the gut microbiota is indeed possible, but protein/peptide/amino acids are also able to modulate a wide range of cellular pathways that can be relevant in the context of inflammation (for instance Tryptophan metabolism pathways).

8) Based on table S1, proteins represent up to 90% of the supplements, what are the remaining 10 percents? Lipids ? Could the authors exclude effects conferring by this fraction?

Minor points:

Line 17: “and suppression of inflammatory bowel disease”: Is there really clinical studies demonstrating the suppression of IBD by dietary interventions? If not this sentence should be edited to be less overstated.

Line 17-18 : “which at a large  extent is attributed to the anti-inflammatory properties of nutrients.” To my point of view, this is an overly simplistic view. Even if some nutrients are considered as anti-inflammatory compounds, vast majority of their effects on immune responses rely on their sensing by metabolic pathways in cells (mTor, AMPK, PI3K, and so on…) and the complex crosstalks of these pathways with immune signaling. This sentence should be edited to take into account this comment.

Line 18-19: “Diets containing fish are known 18 to promote health and suppress inflammatory diseases.” Again, there is a lack of nuance in this sentence; is there really curative effects of diet enriched in fish in the context of inflammatory diseases? If not, please edit.

Line 27 “inflammatory bowel disease” : Replace inflammatory bowel disease (human disease) by colitis or chemically-induced colitis.

Line 28 “To determine the potential mechanism of action”  Pro-inflammatory cytokines/chemokines are good hallmarks of the colitis, informing on the severity of the colitis, but these molecular players do not represent a “mechanism of action” (i.e. signaling pathways linking protein hydrolysates to the expression of these inflammatory mediators, such as mTor maybe). Sentence should be edited (for example: to evaluate colitis intensity/severity)

Line 40 “Intestinal homeostasis is at a large extent maintained by a balance between the host immune system, the gut epithelium, and dietary metabolites” Add gut microbiota as a key element sustaining also intestinal homeostasis.

Table S1 legend: “the cutaneous model.” Paper is about colitis, why here we have “cutaneous”? Please clarify.

Line 108: “indicating that HSH supplement administration suppresses ” Replace “indicating” by suggesting. There is no direct clue here of inflammation suppression.

Line 156-158: “Tissue resident macrophages of a healthy gut exhibit an anti-inflammatory role by producing anti-inflammatory cytokines, particularly IL-10, in order to countebalance inflammatory events and maintain colon homeostasis [27].” I am not convinced that tolerogenic state of the gut associated immune system relies only/mainly on macrophages producing IL-10. Many others immune cells (dendritic cells, intraepithelial lymphocytes, and so on) and immune mediators (IL-4, TGF-beta, …) orchestrate this tolerogenic response. Please rephrase accordingly to take into account this complexity.

Figure S1: It is quite relevant to measure this cytokine at protein level, it is a pity that the result is not in the main figure (could be added to figure 4) and that there is only one inflammatory marker measured at protein level (lipocalin 2 or IL-8 could be of interest to reinforce the findings. It is surprising that the authors were not able to measure detectable TNF-alpha in these settings.

Figure 4C: mention of p=0.019 could be removed.

Author Response

Comment: This study evaluates the immunomodulatory properties of fish-derived proteins in the context of DSS-induced colitis. It is an interesting topic, but there are two main limitations of this study (detailed among the major points below):

DSS colitis is induced by adding DSS in the drinking water, thus severity of the colitis is intimately linked to the amount of water ingested by mice. At two times in the manuscript, the authors hypothesized that maybe their supplements increase salt concentration in mice food, and thus potentially increase their water consumption. If it is the case, we cannot conclude on the results of this study. As mentioned below, it is important that the authors evaluate the effects of their supplements on daily food and water intake, since these parameters have major effects on colitis severity.

Some classical readout of colitis are missing: histological analysis, intestinal permeability assay (dextran FITC) and more than one protein marker for evaluating colitis (only IL-6 in this study). All the study is based on the measure of mRNA level during the recovery period, it is informative but not totally relevant since transcriptional regulation of cytokines/chemokines is a highly dynamic process that might be complex to interpret.

Response: We thank the reviewer for the constructive suggestions. We have performed additional analyses as described below to address the limitations indicated by the reviewer, which we believe that have significantly improved the manuscript.

Major points:

Comment 2.1: Introduction is very short, it might be of interest for the readers to more the already known effects of dietary interventions in the context of IBDs. Notably, are there some studies describing the effects of fish-based supplement (as evoked in the abstract by the authors :” Diets containing fish are known to suppress inflammatory diseases.”and/or based on protein hydrolysates. The paragraph between lines 70 (the fish industry…) and 79 (protein) should be moved in the introduction section (it does not contain results, and proposes interesting introductory elements).

Response 2.1: We thank the reviewer for the useful feedback and the suggestions. The introduction has been expanded by describing dietary interventions in the context of treating IBDs (lines 64-74), as well as the effect of fish-containing diets (lines 80-87). Moreover, the part of the lines 70-79 from the original manuscript has been moved in the introduction in the revised manuscript (lines 88-99), as the reviewer suggested.

Comment 2.2: Line 69: “Fish-derived protein hydrolysates partly suppresses inflammatory bowel disease development.”. Here (and throughout the manuscript) , if the authors refer to their model of DSS-induced colitis, they have to say “suppresses colitis” or “suppresses chemically-induced colitis” or “DSS-induced colitis” but not inflammatory bowel disease that refer to the human disease. This is confusing. Please edit here and throughout the manuscript.

Response 2.2: We thank the reviewer for the comment. The manuscript has been edited as suggested.

Comment 2.3: Diet supplemented with fish-derived products might be consumed differently by mice since they do not have the same organoleptic properties as the control diet. Is there a control or measure of daily food intake between groups? Because, if there are some differences in the daily food intake between groups it can impact the response to colitis (for instance, caloric restriction is associated with anti-inflammatory effects). If available these data should be provided, if not, this possible bias should be mentioned by the authors.

Response 2.3: We thank the reviewer for stressing this important issue. In order to examine whether there are any differences in food intake between the diet groups, we measured daily food intake of mice receiving the different supplements for a week. No significant differences were observed in any of the groups. A new graph with the daily food intake has been added in the supplementary materials as supplementary figure 1B.

Comment 2.4: Line 129-130 and Line 143-144: “or due to the higher salt content of the aforementioned supplements which could lead mice to a slightly higher DSS-containing water consumption” It might represent an important bias, ideally daily water drinking should be measured to evaluate the importance of this bias (alternatively another colitis model that not relies on drinking water should be used (TNBS for instance)).

Response 2.4: This matter is of great importance, we thank the reviewer for emphasizing it. As described above (Response 2.3), in addition to food intake, daily measurements of water consumption were also taken for a week for mice receiving the different supplements. Only the HMH group exhibited a statistically significant higher water consumption compared to the control group, whereas none of the rest of the groups did. The higher water intake of the HMH group might explain its higher levels of some of the pro-inflammatory markers examined in the paper. A new graph depicting the daily water consumption of each diet group has been introduced as supplementary figure 1A and the discussion of the manuscript has been edited accordingly (lines 163-171) to include this information. Also, in light of the new data we have removed the hypothetical statements on water consumption.

Comment 2.5: Scoring system used in figure 3 is quite superficial and not very discriminant between groups, it is a pity that there is no histological scoring performed on paraffin embedded colon + HE staining (as usually performed in this kind of experiment).

Response 2.5: We thank the reviewer for the insightful comment. A histological analysis would indeed be more informative. Since we had stored intestinal samples for that exact purpose, we performed an H&E histological analysis. The results have been added as figure 1D and text has been edited accordingly.

Comment 2.6: Except for IL-6 (presented in figure S1) all colitis hallmarks were measured at transcriptional level. Since these measures have been done during the recovery period and only based on transcripts, it might be complex to conclude on the results obtained.

Response 2.6: We thank the reviewer for the comment. In order to support our data we moved the data of the elisa experiments of figure S1 to the main text as figure 6. Also elisa of cxcl1 (KC/IL8) was performed in mouse serum, which has also been added as figure 6C. In colon tissue cxcl1 protein was undetectable.

Comment 2.7: As noticed for the introduction, the (final) discussion section is very short, too. It might be of interest to develop a bit more on the potential mechanism involved to explain the immunomodulatory properties of these protein hydrolysates. Effects on the gut microbiota are indeed possible, but protein/peptide/amino acids are also able to modulate a wide range of cellular pathways that can be relevant in the context of inflammation (for instance Tryptophan metabolism pathways).

Response 2.7: We thank the reviewer for the interesting suggestion. The discussion section has been expanded by examining the possibility that the tested protein hydrolysates could act through regulation of metabolic pathways, particularly amino acid pathways, besides gut microbiome modulation (lines 332-473) as follows: “The exact mechanism through which the examined supplements act in the above settings of acute intestinal inflammation would be immensely difficult to be defined and beyond the scope of this study. The former is justified by the tremendously complex interplay between dietary components, host immunity, the intestinal barrier function, as well as the gut microbiota. It is highly likely that the tested protein hydrolysates of fish origin modulated the composition of the gut microbiome, considering that animals were exposed to the supplements for a total of 15 days (7 days prior to the experiment plus 8 days of the experiment) and the fact that intestinal microbiota may respond promptly to modified diets [59]. In agreement with our hypothesis, tuna-derived fish supplements were shown to ameliorate DSS-induced intestinal inflammation through short-chain fatty acid production and microbiome modulation [60]. We have also recently shown that different protein-containing dietary supplements affect the gut microbiome.

The tested supplements consist primarily of protein but also a fraction of minerals. Lipids were actively removed and a remnant of 1-2% was present, thus their potential contribution in the anti-inflammatory action is rather limited. The contribution of minerals cannot be excluded, since minerals such as Zn possess anti-inflammatory actions. Nevertheless the major content of the supplements are proteins consisting of a wide range of amino acids. It has been exhibited that intestinal inflammation alters amino acid metabolism both in the immune and non-immune cells of IBD patients and their intestinal microbiome, thus creating a shift in nutritional demands and utilization [61,62]. Possibly, each amino acid has a distinct part in intestinal inflammation, with some exacerbating it and others suppressing it [16,63,64]. For example, dietary glycine protected rats from chemically-induced colitis by abrogating the production of pro-inflammatory cytokines and chemokines [65]. Interestingly, the HSH supplement, which had the most beneficial effect of the examined extracts, is rich in glycine, as are the HSB and HMB extracts, which also presented some anti-inflammatory action (Table S1). Notably, collagen has the highest glycine concentration, yet it presented an increase of both anti-inflammatory and pro-inflammatory markers; thereby it would be over-simplistic to attribute the observed effects solely to a specific amino acid. Instead, the entire amino acid profile of each extract and the potential action of bioactive peptides should be accounted for.

Moreover, DSS-induced colitis in rats was alleviated when supplemented with threonine, serine, proline and cysteine, in part through stimulation of mucin production and equilibration of the gut microbiome [66]. With the exception of cysteine, the HSH, HSB and HMB supplements contain considerable levels of the aforementioned amino acids. Histidine downregulated the production of pro-inflammatory markers by macrophages when offered in mice with colitis [67]. Arginine, which is found in important concentrations in the examined fish extracts, reduced intestinal inflammation in rats and mice with chemically-induced colitis [68-70]. Overall, the fish-derived protein hydrolysates that were tested in this study contain considerable quantities of amino acids that have been demonstrated to ameliorate symptoms of colitis, through modulation of amino acid metabolic pathways.”

Comment 2.8: Based on table S1, proteins represent up to 90% of the supplements, what are the remaining 10 percents? Lipids ? Could the authors exclude effects conferring by this fraction?

Response 2.8: We thank the reviewer for the comment. From the preparation of the diets lipids were actively removed and only a maximum 1-2% of lipid remnants were present in the supplements. Thus, the remaining percentage of the diets consisted of minerals, which were not removed because free amino acids would have been also removed in that case. The potential action of minerals cannot be excluded and it is now discussed in the manuscript as follows:” The tested supplements consist primarily of protein but also a fraction of minerals. Lipids were actively removed and a remnant of 1-2% was present, thus their potential contribution in the anti-inflammatory action is rather limited. The contribution of minerals cannot be excluded, since minerals such as Zn possess anti-inflammatory actions. Nevertheless the major content of the supplements are proteins consisting of a wide range of amino acids.”

Minor points:

Comment 2.9: Line 17: “and suppression of inflammatory bowel disease”: Is there really clinical studies demonstrating the suppression of IBD by dietary interventions? If not this sentence should be edited to be less overstated.

Response 2.9: We thank the reviewer for the comment. There is growing evidence from both basic and clinical research studies that diet could increase or lower the risk of IBD and that dietary interventions could alter risk or disease development. Some recent review papers which summarize the above are PMID: 27793606 and PMID: 33801883. The introduction has been updated to include dietary interventions in the context of colitis, as suggested in comment 2.1.

Comment 2.10: Line 17-18 : “which at a large  extent is attributed to the anti-inflammatory properties of nutrients.” To my point of view, this is an overly simplistic view. Even if some nutrients are considered as anti-inflammatory compounds, vast majority of their effects on immune responses rely on their sensing by metabolic pathways in cells (mTor, AMPK, PI3K, and so on…) and the complex crosstalks of these pathways with immune signaling. This sentence should be edited to take into account this comment.

Response 2.10: We thank the reviewer for the feedback. “the anti-inflammatory properties of nutrients” has been changed to “the complex immunomodulatory properties of nutrients”, to avoid making an over-simplistic statement. But since we do not explore the underlying molecular mechanism of action in this study, we avoided expanding on the possibly involved molecular pathways in the abstract.

Comment 2.11: Line 18-19: “Diets containing fish are known to promote health and suppress inflammatory diseases.” Again, there is a lack of nuance in this sentence; are there really curative effects of diet enriched in fish in the context of inflammatory diseases? If not, please edit.

Response 2.11: We thank the reviewer for the comment. Please refer to response 2.9. Diets containing fish, especially omega 3 fatty acids, have been exhibited to reduce inflammation in inflammatory disease in murine models. Relative references have been added in the introduction. The current evidence on humans is still inconclusive, but some beneficial trends can be noted (please see for example PMID: 31574900, PMID: 30680455). Taking your comment into consideration, in the above lines “are known to” has been edited to “have been suggested to”, in order to make a less conclusive statement.

Comment 2.12: Line 27 “inflammatory bowel disease” : Replace inflammatory bowel disease (human disease) by colitis or chemically-induced colitis.

Response 2.12: Manuscript has been corrected as suggested.

Comment 2.13: Line 28 “To determine the potential mechanism of action”  Pro-inflammatory cytokines/chemokines are good hallmarks of the colitis, informing on the severity of the colitis, but these molecular players do not represent a “mechanism of action” (i.e. signaling pathways linking protein hydrolysates to the expression of these inflammatory mediators, such as mTor maybe). Sentence should be edited (for example: to evaluate colitis intensity/severity)

Response 2.13: “To determine the potential mechanism of action” was edited to “To evaluate colitis severity”. Thank you for the comment.

Comment 2.14: Line 40 “Intestinal homeostasis is at a large extent maintained by a balance between the host immune system, the gut epithelium, and dietary metabolites”. Add gut microbiota as a key element sustaining also intestinal homeostasis.

Response 2.14: Line 40 was edited to include the gut microbiome, thank you.

Comment 2.15: Table S1 legend: “the cutaneous model.” Paper is about colitis, why here we have “cutaneous”? Please clarify.

Response 2.15: “cutaneous model” has been corrected toDSS-induced colitis model”. Thank you for pointing this out.

Comment 2.16: Line 108: “indicating that HSH supplement administration suppresses ” Replace “indicating” by suggesting. There is no direct clue here of inflammation suppression.

Response 2.16: “indicating” has been replaced by “suggesting”, thank you for the comment.

Comment 2.17: Line 156-158: “Tissue resident macrophages of a healthy gut exhibit an anti-inflammatory role by producing anti-inflammatory cytokines, particularly IL-10, in order to counterbalance inflammatory events and maintain colon homeostasis [27].” I am not convinced that the tolerogenic state of the gut associated immune system relies only/mainly on macrophages producing IL-10. Many other immune cells (dendritic cells, intraepithelial lymphocytes, and so on) and immune mediators (IL-4, TGF-beta, …) orchestrate this tolerogenic response. Please rephrase accordingly to take into account this complexity.

Response 2.17: We thank you for the comment. The above lines have been edited in such a way to express more of a suggestive than explicit statement (including a new reference), and an introductory sentence was placed prior to them, noting the complexity of the gut tolerogenic state (lines 209-215).

Comment 2.18: Figure S1: It is quite relevant to measure this cytokine at protein level, it is a pity that the result is not in the main figure (could be added to figure 4) and that there is only one inflammatory marker measured at protein level (lipocalin 2 or IL-8 could be of interest to reinforce the findings. It is surprising that the authors were not able to measure detectable TNF-alpha in these settings.

Response 2.18: We thank the reviewer for the suggestion. As mentioned in response 2.6, the data of figure S1 have been moved to the main text and we have now measured serum levels of IL8(cxcl1), which is also included in revised Figure 6.

Comment 2.19:Figure 4C: mention of p=0.019 could be removed.

Response 2.19: We thank the reviewer for the comment, the figure has been corrected.

Round 2

Reviewer 1 Report

Manuscript is improved and should be accepted.

Author Response

We thank the reviewer for the positive view of our work.

Reviewer 2 Report

The authors have addressed almost all my concerns. Thank you. The manuscript is significantly improved to my point of view.

A last thing that can be improved is to score the histological damages of figure 1D (something that can be performed very quickly by a pathologist by scoring histological damages : for instance by following such guideline: "Histologic scoring: Blinded histologic scoring can be performed on H & E stained colonic tissue as follows. Each section is assigned four scores based on the degree of epithelial damage and inflammatory infiltration into the mucosa, submucosa and muscularis/serosa. Each of the four scores is multiplied by 1 if the change was focal, 2 if it was patchy and 3 if it was diffuse. The 4 individual scores per colon are added, resulting in a total scoring range of 0–36 per mouse. The average scores for control and DSS-treated groups can then be tabulated." (https://www.ncbi.nlm.nih.gov/pmc/articles/PMC3980572/)

It will give to the authors a histological activity index (HAI) that is usually displayed along HE stainings micrographs to quantify the extent of damages.

Author Response

Comment: A last thing that can be improved is to score the histological damages of figure 1D (something that can be performed very quickly by a pathologist by scoring histological damages : for instance by following such guideline: "Histologic scoring: Blinded histologic scoring can be performed on H & E stained colonic tissue as follows. Each section is assigned four scores based on the degree of epithelial damage and inflammatory infiltration into the mucosa, submucosa and muscularis/serosa. Each of the four scores is multiplied by 1 if the change was focal, 2 if it was patchy and 3 if it was diffuse. The 4 individual scores per colon are added, resulting in a total scoring range of 0–36 per mouse. The average scores for control and DSS-treated groups can then be tabulated." (https://www.ncbi.nlm.nih.gov/pmc/articles/PMC3980572/). It will give to the authors a histological activity index (HAI) that is usually displayed along HE staining micrographs to quantify the extent of damages.

Response: We thank the reviewer for suggesting this scoring and providing relevant literature. Blinded histological scoring has been performed according to the suggested protocol and has been added as figure 1E. Scoring is described as follows: “In order to further assess DSS-induced tissue damage, blinded histological scoring was performed according to Chassaing et al. [38](Figure 1E). In accordance with the colon length measurements, HSH supplementation reduced DSS-induced tissue damage and in addition HSB and HSH+C significantly improved colon histological features post DSS treatment (Figure 1E)” (Lines 138-142 and 440-443).